# Charge carrier-selective contacts for nanowire solar cells

Sebastian Z. Oener[1,2], Alessandro Cavalli[3], Hongyu Sun[2], Jos E.M. Haverkort [3], Erik P.A.M. Bakkers [3,4] & Erik C. Garnett [2]

Charge carrier-selective contacts transform a light-absorbing semiconductor into a photo-voltaic device. Current record efficiency solar cells nearly all use advanced heterojunction contacts that simultaneously provide carrier selectivity and contact passivation. One remaining challenge with heterojunction contacts is the tradeoff between better carrier selectivity/contact passivation (thicker layers) and better carrier extraction (thinner layers). Here we demonstrate that the nanowire geometry can remove this tradeoff by utilizing a permanent local gate (molybdenum oxide surface layer) to control the carrier selectivity of an adjacent ohmic metal contact. We show an open-circuit voltage increase for single indium phosphide nanowire solar cells by up to 335 mV, ultimately reaching 835 mV, and a reduction in open-circuit voltage spread from 303 to 105 mV after application of the surface gate. Importantly, reference experiments show that the carriers are not extracted via the molybdenum oxide but the ohmic metal contacts at the wire ends.

[1] Department of Chemistry and Biochemistry, University of Oregon, Eugene, OR 97403, USA. [2] Center for Nanophotonics, AMOLF, Science Park 104, 1098 XG Amsterdam, Netherlands. [3] Applied Physics, Eindhoven University of Technology, PO Box 513, 5600 MB Eindhoven, Netherlands. [4] Kavli Institute of Nanoscience, Delft University of Technology, Delft 2629HZ, Netherlands. Correspondence and requests for materials should be addressed to S.Z.O. (email: szo@uoregon.edu) or to E.C.G. (email: garnett@amolf.nl)

Carrier-selective contacts are an essential component of solar cells[1]. Traditionally, the semiconductor is doped with specific impurity atoms, raising the electron or hole concentration and hence conductivity in the highly doped regions. However, those regions suffer from increased nonradiative recombination (especially Auger-type), decreased carrier mobility and parasitic light absorption that increases minority carrier conductivity and decreases contact selectivity[1]. Therefore, high-efficiency wafer-based silicon[2–10], III-V[11,12] (https://www.altadevices.com/technology/), organic[13,14], and perovskite[15–18] solar cells all employ heterojunction contacts providing simultaneously a high degree of carrier selectivity in the adjacent large band gap material and excellent interface passivation. One difficulty that arises with heterojunction contacts is that there is often a tradeoff between the need for thicker layers to improve contact selectivity and reduce contact recombination, while needing thin layers to minimize carrier extraction losses (especially at high current density) and parasitic absorption. Even for the 5−10-nm-thin a-Si:H(i) layers that are used to passivate current high-efficiency silicon heterojunction solar cells, a thickness increase, if allowed by removing electrical and optical limitations[19], could still lead to an improved open-circuit voltage ($V_{OC}$) from currently 738 to 750 mV[20–22] to the Auger limit of about 760 mV (for 110 μm wafer thickness)[23] due to strongly thickness-dependent passivation properties for films below approx. 15 nm[24,25]. Those limitations are also present for the heterojunction interfaces that have been realized for nanowire solar cells[26–30]. However, nanowire photovoltaics can in principle decouple the carrier selectivity and extraction functions of the heterojunction by using the extreme surface sensitivity to control electron and hole concentrations in the vicinity of the contact. Such an approach is commonly used in electronics where an electrostatic gate voltage can drastically alter the carrier concentration in a narrow surface channel adjacent to electrical contacts, causing accumulation, depletion or even inversion without the need for an interfacial layer in between the contact and channel. Nanowires with a wrap-around gate structure provide the ideal geometry for maximum gate coupling and have already shown excellent performance[31–36]. It is also possible to remove the need for a gate voltage by employing surface layers that either donate or withdraw electric charge to act as a permanent fixed gate[37–40]. Taking this one step further, such a surface layer has been used to create a nanowire solar cell without doping[28,41]. Even though the performance was somewhat worse than that of state-of-the-art nanowire solar cells, the approach is very appealing due to the difficulty of controlled doping at the nanoscale[42] and the excellent surface gate coupling.

Here, we show how the strong surface sensitivity of InP nanowires can be used to alter the charge carrier selectivity of the hole contact while keeping the extraction path unchanged. One of the main challenges of InP nanowires, in fact III-V semiconductor materials in general, is the formation of the hole-selective contact, that is traditionally the highly p-type doped region. The most widely used and also here employed p-dopant Zn shows strongly limited incorporation dynamics during growth, strongly increasing diffusion constants with increasing concentration[43–45] and is known to even cause increased nonradiative recombination[46]. Therefore, the formation of short (smaller than 500 nm) highly doped p-type nanowire segments with an abrupt doping profile is very challenging[47,48]. For InP nanowires those problems are even amplified due to Fermi level pinning under the conduction band caused by the native oxide. It is because of those reasons that we focus here on the hole contact while our approach is also applicable to the electron contact, given the successful realization of dopant-free n-type heterojunction contacts in bulk solar cells in the past[3,5,11,49]. First, we fabricate ohmic contacts to a horizontal InP $p$-$i$-$n$ junction nanowire solar cell followed by selective surface modification next to (not underneath) the hole contact. Removing the native oxide by HF etching and depositing $MoO_X$ increases the $V_{OC}$ by up to 335 mV, reaching values up to 835 mV. This $V_{OC}$ value is comparable to that obtained for record single InP nanowires (800 −890 mV)[50,51] and nanowire array InP solar cells (760 mV (17.8% efficiency) and 906 mV (13% efficiency))[52,53] and even close to that of record bulk InP solar cells (currently 939 mV)[54], which is quite remarkable given no special surface passivation has been applied[55]. Our results demonstrate that the nanowire geometry allows for a traditional heterojunction layer to act as a surface gate, increasing the local hole concentration and thereby providing excellent carrier selectivity by changing the effective doping concentration, without changing the impurity doping level at the contact. In contrast to traditional heterojunction contacts, the surface gate approach does not require conduction through the often resistive heterojunction contact material itself, making it possible to use very thick surface gate layers without introducing a charge carrier extraction barrier.

## Results

**Schematic overview**. Figure 1 schematically shows three different types of charge carrier-selective contacts: traditional doped semiconductor homojunction, traditional heterojunction, and nanowire surface gating contacts. For traditional homojunction solar cells (Fig. 1a), charge carrier selectivity of a contact is induced by doping the underlying semiconductor region with impurities, thereby increasing the carrier density and conductivity of one charge carrier while decreasing the conductivity for the opposite charge carrier. Traditional heterojunction contacts rely instead primarily on local accumulation/inversion (change in carrier density but not impurity doping level) inside the semiconductor caused by a difference in Fermi level at the interface to establish carrier-selective contacts (Fig. 1b). Importantly, the heterojunction interface is required to be free of charge carrier extraction barriers and to provide asymmetric band offsets for electrons and holes. Furthermore, once a suitable heterojunction contact material is found it often has to be kept thin to limit resistance and absorption losses. In this study, we show that nanowires allow for another type of charge carrier-selective contact, which can strongly reduce the requirements compared to traditional heterojunction interfaces. Nanowires can utilize surface gating layers, such as high or low work function oxides, to induce carrier accumulation/inversion in the semiconductor, i.e. they control the local carrier concentration without changing the doping level (Fig. 1c). However, in stark contrast to traditional solar cells, the charge carriers can be extracted via ohmic metal point contacts at the nanowire end segments; the employed surface layers are not being used for extraction but instead act like a local chemical gate to induce the required selectivity (see also cross-sectional image in Fig. 1c). Therefore, the requirements are strongly reduced compared to traditional heterojunction interfaces.

**Influence of surface gate on open-circuit voltage**. To study this nanowire surface gating selective contact, we use single horizontal $p$-$i$-$n$ junction InP nanowires grown by selective area epitaxy[56,57] (200 nm diameter with a 50 nm $SiO_2$ shell) with ohmic contacts at the ends. The $SiO_2$ shell was used to increase long-term stability, while also avoiding clustering into nanowire bundles during the drop casting on the electrode chips. By using single nanowire devices, the impact of different surface treatments can be studied directly via $I$−$V$ characteristics and unobscured by average effects over millions of wires on the typical nanowire array level (see

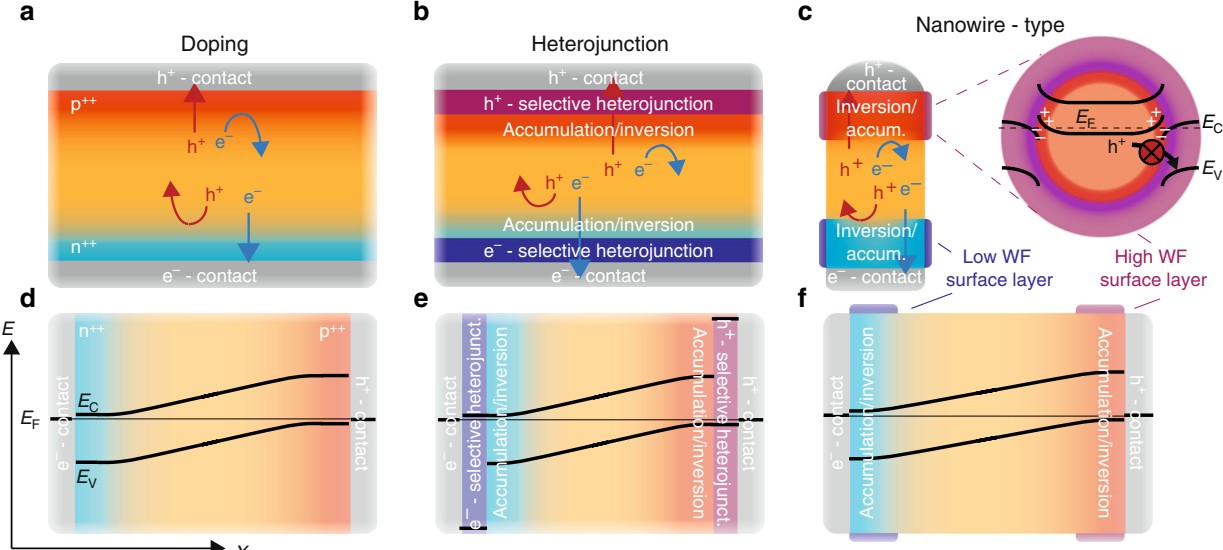

**Fig. 1** Different types of charge carrier-selective contacts. **a** Contact selectivity of traditional solar cells is determined by the doping density of the semiconductor directly underneath the metal contact. **b** Contact selectivity of heterojunction solar cells is determined by the induced accumulation/inversion region inside the semiconductor directly underneath the metal contact, due to the work function difference between heterojunction contact material and semiconductor. **c** Nanowires allow for a different type of charge carrier-selective contact; the carriers can be extracted parallel to the surface instead of perpendicular to it (as in doped- and heterojunction contacts) (red and blue arrows). This means that carriers are not extracted through the surface-gate layer. The cross-sectional image shows a possible band alignment and the blocking of radial hole transport at the surface-gate interface. In **d**–**f**, the band diagrams in the dark are drawn for the cases in **a**–**c**, respectively. We note that band bending in the dark is indicative for the selectivity of a contact but not a sufficient description. For an accurate assessment of charge carrier selectivity the quasi-Fermi level under illumination have to be considered, which have been omitted here for simplicity. The color gradient indicates the charge carrier selectivity with red being very hole selective and blue very selective for electron conduction

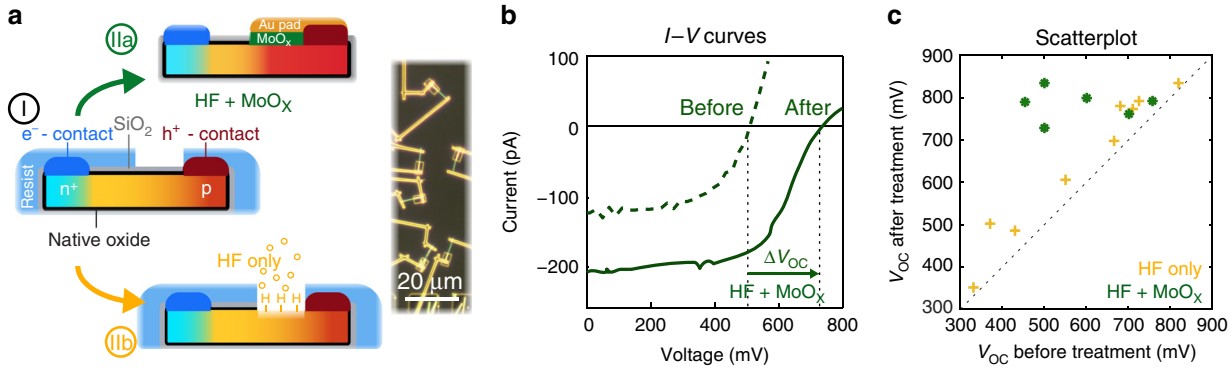

**Fig. 2** Improving carrier selectivity with $MoO_X$. **a** Schematic of experimental setup. Contacted single InP nanowire $p$-$i$-$n$ junction solar cells are coated with electron-beam resist. A window is opened in the resist next to the hole contact to test the device characteristics before any treatment (I), after HF (IIb) or after HF and 30 nm $MoO_X$ layer evaporation (IIa). To prevent degradation of the $MoO_X$ work function due to ambient contaminants, a 100 nm Au capping layer (orange) has been evaporated. The right side shows an dark-field optical microscope image of single nanowire solar cells before surface treatment, where the resist windows are visible. **b** $I$–$V$ curve before and after HF + $MoO_X$ treatment. The vertical dashed lines indicate the open-circuit voltage. **c** Scatter plot showing each single nanowire solar cell $V_{OC}$ before and after HF etching (yellow) or HF etching + $MoO_X$ evaporation (green)

optical microscope image in Fig. 2a). After the ohmic contact formation we coat the entire device with electron beam resist and open windows along the p-type segment of the nanowire next to the ohmic metal contact where we apply different surface treatments to induce the surface gate (I, IIa, and IIb in Fig. 2a). Previously, we measured a large difference between the optically implied $V_{OC}$[58,59] and the electrically extracted $V_{OC}$ using similar InP nanowire solar cells indicating that the electrically extracted $V_{OC}$ is limited by contact selectivity for those devices and not by surface or bulk recombination[51].

Figure 2b shows the $I$–$V$ curve before (dashed) and after (solid) the high work function interfacial layer $MoO_X$ has been

evaporated onto the same device after HF etching (approx. 10 min exposure to air). The work function of $MoO_X$ has been determined to be between about 5.7 eV and 6.6 eV, depending on the amount of carbon contamination (decreasing with increasing carbon content)[9]. For wurtzite InP the valence band lies around 5.75 eV while the band gap is 1.43 eV (300 K)[60], explaining the ability of $MoO_X$ to create a hole accumulation region inside the semiconductor. The increase in $V_{OC}$ is clearly visible for the depicted device; the $V_{OC}$ increases by 230 mV, reaching 730 mV. All the single InP nanowire solar cells, for which we measured the same device before and after the surface treatment, showed a substantial increase in $V_{OC}$ (Fig. 2c). Interestingly,

although there was a very large spread in $V_{OC}$ of the devices relying only on the p-i-n junction doping for the carrier selectivity (454−757 mV), after $MoO_X$ surface coating all the devices showed high $V_{OC}$ values (730−835 mV) and no remaining correlation with the original $V_{OC}$. This suggests that doping nonuniformities were causing nearly all the variation in $V_{OC}$ observed from wire to wire and that $MoO_X$ surface coatings are capable of fixing poor carrier selectivity by inducing a higher hole concentration via a surface gate effect. Photoluminescence images before the contact formation indeed indicate doping nonuniformities of the as-grown wires (see Supplementary Figure 1 and Supplementary Note 1).

To further support the surface gate hypothesis we also fabricated device geometries where the $MoO_X$ pad covered the n-type part of our nanowire solar cells. This device geometry resulted in strongly decreased performance (Supplementary Figure 2a). Furthermore, experiments on symmetrically doped p-type wires and a $MoO_X$ pad covering the central nanowire part showed an increase in conductivity (Supplementary Figure 2b).

Therefore, we can clearly state that the $MoO_X$ increases the hole-conductivity and hence selectivity of the p-type part in Fig. 2. Interestingly, several nanowire devices that had a small gap between the surface gate window and original metal contact (due to misalignment in the last lithography step) still showed large $V_{OC}$ improvement (representative example in Supplementary Figure 3). This suggests that the $MoO_X$ is not acting as a traditional heterojunction, where carriers are extracted via the heterojunction layer, but instead only as a surface gate, with carriers extracted directly via the metal contact.

Figure 2b shows the occurrence of an s-shaped $I−V$ curve after the surface treatment, while Supplementary Figure 3 clearly shows the s-shaped character already before the treatment. We ascribe the observed extraction barriers to nonideal effective doping concentrations along the hole extraction path (for a detailed discussion see SI). This is further supported by the resistive behavior of our devices in the dark, as shown in Supplementary Figure 4.

While the increase in $V_{OC}$ with $MoO_X$ surface modification is consistent across all measured devices, the short-circuit current $I_{SC}$ increases for some devices (e.g. Fig. 2b) or decreases (e.g. Supplementary Figure 3) after the treatment. Importantly, the increase in $I_{SC}$ is only observed for devices with relatively poor initial performance, while for initially good performing devices, the $I_{SC}$ always decreases. The decrease can be explained by the opaque Au coverage of the $MoO_X$, which is used in order to maintain the high $MoO_X$ work function and avoid its degradation due to ambient contaminants.

We note that our current results are strongly limited by the horizontal single nanowire device geometry. A wrap-around gate geometry with an ohmic metal contact only at the very tip of a short nanowire can not only remove the observed s-shaped character but also increase hole-selectivity to even higher values, due to more uniform surface gate coupling. Currently, the nanowire region next to the surface facing the substrate will have a smaller change in carrier concentration due to the asymmetric deposition of $MoO_X$ in this proof-of-concept geometry.

Motivated by those results, we also fabricated single nanowire devices with the traditional interfacial layer geometry where $MoO_X$ (15 nm) is present everywhere between the p-type InP and the Au contact. However, all of the fabricated devices (ca. 50) showed very high resistance or no apparent electrical contact at all. Therefore, we can conclude that the interfacial layer on our devices shown here improved the carrier selectivity indeed without changing the extraction path, as was already indicated by the observation of $V_{OC}$ improvements despite unintended gaps between metal contacts and $MoO_X$ pads; even though $MoO_X$ is

present on the surface, the charge carriers are still being extracted via the ohmic contact at the nanowire end. This observation indicates a large charge carrier extraction barrier at the InP| $MoO_X$ interface, which we speculate is related to negative charging of the $MoO_X$ layer that is the origin of the upward band bending in the InP (causing hole accumulation) and also causes downward band bending in the $MoO_X$ (creating a hole extraction barrier) (see Supplementary Figure 5 and Supplementary Notes 2 and 4). This observation is different from other semiconductor interfaces where $MoO_X$ has been shown to essentially act as a high work function metal (5.75−6.70 eV) which establishes a selective hole contact[9].

**The role of hydrogen fluoride etching**. To understand this enhancement in $V_{OC}$ better and isolate the influence of surface recombination in the p-type nanowire region (traditionally part of contact recombination) on the improved contact selectivity,[61] we also studied nanowire devices that have only been exposed to HF, that is without the subsequent $MoO_X$ evaporation. Nanowires are especially prone to surface recombination which can directly reduce the sustained carrier concentration under illumination and hence the $V_{OC}$. HF serves as a benchmark surface passivation treatment for InP, because of its ability to completely remove the native oxide[62], and therefore the HF-only treatment and rapid subsequent characterization (less than 1 min exposure to air) can be used to investigate the effect of surface passivation without the presence of the $MoO_X$ chemical gate. Figure 2c shows that HF treatment did indeed lead to a small and consistent increase in $V_{OC}$ for all devices. The $I−V$ curve of one representative device is shown in Fig. 3a. However, unlike the case of $MoO_X$ surface deposition, HF treatment did not remove the large variation in starting $V_{OC}$ value, suggesting that it only passivates the surface and does not improve the carrier selectivity (or only slightly). Figure 3b shows the effect of the native oxide more clearly; immediately after HF etching (less than 1 min exposure to air) the $V_{OC}$ improves, but over time in ambient air it decreases again due to reoxidation. Most of the surface passivation effect is lost even after only 30 min and by 20 h the $V_{OC}$ has returned to its original value. The HF treatment in Fig. 3 therefore has a clearly distinct effect on the nanowires compared to the $MoO_X$ + HF treatment in Fig. 2. It increases the $V_{OC}$ by smaller values and it does not remove the initially large spread in $V_{OC}$ even though the wires have been exposed to the ambient for a shorter time (about 1 min) than for the $MoO_X$ + HF treatment (about 10 min) (see Methods). These results coupled with the initially highly inhomogeneous PL between and within nanowires (see Supplementary Figure 1 and Supplementary Note 1) suggest that changes in carrier concentration, rather than surface passivation, explain the large improvements caused by the $MoO_X$ surface layer deposition. Nonetheless, there is still the possibility that the initial surface passivation varies from wire to wire and even along individual wires, causing all the observed effects. However, our results for devices for which $MoO_X$ surface layers were deposited on the n-type side of the solar cells showed a strongly decreased performance (Supplementary Figure 2a). Together with our results in Fig. 2 and the increased conductivity for $MoO_X$ pads located on the center of symmetrically doped p-i-p wires (Supplementary Figure 2b) we show that surface gating, rather than simple surface passivation, is primarily responsible for the large increases in $V_{OC}$ and uniformity seen with $MoO_X$ deposition.

## Discussion

We have demonstrated a contact geometry where surface layers traditionally used as heterojunctions can be placed next to, instead of underneath, the metal contact to improve carrier

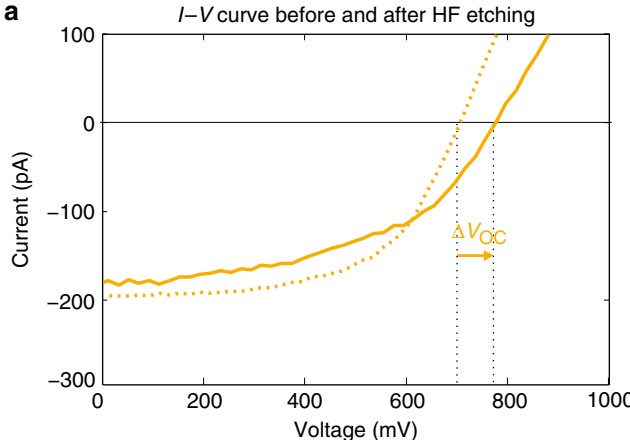

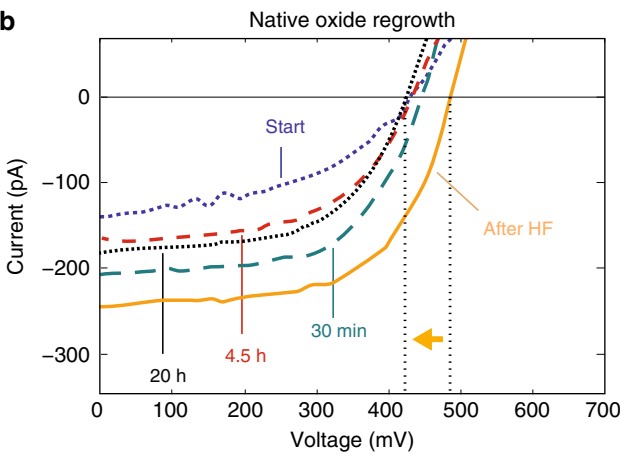

**Fig. 3** Effect of HF etching on the $V_{OC}$. **a** I–V curve before (dashed line) and after (solid line) HF etching. **b** The I–V curves of a second sample show the effect of the native oxide regrowth. After the initial increase in performance (yellow), the native oxide slowly grows back (cyan, red, black), reducing the $V_{OC}$. The vertical dashed lines are guides for the eye

selectivity in nanowire solar cells. The high surface sensitivity in nanowires allows $MoO_X$ surface layers to act as a local permanent gate, leading to a $p^{++}$-type accumulation layer in the underlying InP nanowire and hence increasing the effective doping concentration, which is the mechanism for the increased contact selectivity and $V_{OC}$. Devices with $MoO_X$ underneath instead of next to the contact lead to large charge carrier extraction barriers, proving that the $V_{OC}$ improvements we observe can be ascribed to increased carrier concentration inside the semiconductor, induced by the high work function of the adjacent $MoO_X$, instead of carrier-selective conduction inside the $MoO_X$ itself. Our control experiments also show that surface passivation cannot explain the improved performance we observe.

Our results vividly demonstrate an exciting possibility in nanowire solar cells that does not exist in standard bulk or thin-film geometries: an interfacial layer can be used to improve charge carrier selectivity without the requirement of charge carrier extraction; interfaces free of charge carrier extraction barriers and high conductivity in the heterojunction contact material are not required. This removes the traditional tradeoff between wanting thicker heterojunction layers for better carrier selectivity or contact passivation and wanting thinner heterojunction layers for better carrier extraction. One area where this is particularly relevant is for the emerging class of passivating tunnel contacts[7,8,63,64]. Here an insulating layer is covered by a high or low work function material to provide both carrier selectivity and

passivation. Because carriers must tunnel through the insulating layer, its thickness is limited to about 2 nm, setting very challenging requirements on deposition uniformity, control and interfacial quality and removing the possibility of using thick field-effect passivation layers that have proven so valuable in high-efficiency crystalline silicon solar cells. Nanowire surface gating contacts, especially in wrap-around geometries, open the possibility of using any surface passivation scheme also at the contact. It is important to note that such a scheme can improve carrier selectivity and device uniformity dramatically, especially if accurate control over the doping density and profile is challenging, as shown here and often observed for nanoscale systems.

## Methods

**Sample fabrication.** As described previously[51], the nanowires are grown at a low pressure in an Aixtron 200/4 metalorganic vapor phase epitaxy (MOVPE) reactor via selective area MOVPE (SA-MOVPE) growth. To define the selective growth areas, a 50-nm-thick silicon nitride layer is used as masking layer. The nitride layer is patterned by soft contact nanoimprint lithography. The underlying substrate is a (111)A oriented p-doped InP wafer with a nominal Zn doping carrier concentration of $2\times10^{18}$ $cm^{-3}$ from AXT, USA[65,66]. $H_2$ is used as carrier gas for the precursors, with a total flow of 15 L $min^{-1}$. The growth is performed at a temperature of 730 °C and a pressure of 100 mbar. The precursor gases are trimethylindium (TMI) and phosphine ($PH_3$) with molar fractions $x_i$ (TMI) = $4.7\times10^{-5}$ and $x_i$ ($PH_3$) = $3.9\times10^{-3}$, resulting in a V/III ratio of 83. The total growth time is 11 min and the nominal doping profile is $p^{++}/p/i/n/n^{++}$, with the respective segment growth times of 0.5 min/3 min/4 min/3 min/0.5 min (1 μm/3 μm/4 μm/3 μm/1 μm). The p-type dopant is diethylzinc (DEZn), with molar fractions of $1.3\times10^{-5}$ in the $p^{++}$-region and $6.4\times10^{-6}$ in the p-region. The n-type dopant is ditertbutylsilane (DTBSi), with molar fractions of $9.5\times10^{-6}$ in the $n^{++}$-region and $4.9\times10^{-7}$ in the n-region, respectively. After the growth of the 200 nm diameter wires (±10 nm), a conformal 50-nm-thick $SiO_2$ shell is grown at 300 °C by plasma-enhanced chemical vapor deposition, with the precursors silane and nitrous oxide. The nanowire had a total diameter of 300 nm, a length of 12 μm and $SiO_2$ shell thickness of 50 nm. The InP nanowires form the wurtzite crystal structure, compared to the zincblende crystal structure of bulk wafers. The SA-MOVPE method allows selective semiconductor growth on exposed substrate areas[56,57]. To grow nanowire structures, as done in this work, the growth has to be selective for the top [111]A surface over the {110} side surfaces. For 730 °C and a pitch smaller than 1000 nm we observed negligible growth on the side facets when studied in a TEM. We note that to our knowledge, a detailed theoretical description of the growth mechanism depending on pitch, nanowire diameter, length, dopants and other parameters is currently lacking. Importantly, our results indicate the absence of any pronounced effect of a possible core-shell structure. The nanowires show rectifying photovoltaic behavior as expected for an p-i-n structure when contacted with the positive pole on the p-type side and the negative one on the n-type side. Under illumination, the holes (electrons) are driven towards the p-type (n-type) side, resulting in a negative photocurrent (e.g. Figure 2). Furthermore, hydrogen fluoride (HF) not only etches the native oxide but also InP itself. Therefore, even if a thin (1 to 2 nm) shell exists initially, the HF treatment for the HF-only and the HF + $MoO_X$ + Au treatment is likely to remove this shell. Additionally, the HF treatment is the same for the HF-only and the HF + $MoO_X$ + Au treatment. Therefore, the effect of the $MoO_X$ is still clearly distinct over the HF treatment alone. Last but not least, the results of our photoluminescence measurements (Supplementary Figure 1), the application of $MoO_X$ on the n-type part of p-i-n wires (Supplementary Figure 2a) and symmetric $p^+$-p-$p^+$ wires (Supplementary Figure 2b) are consistent with the assumption of an axial doping geometry, too.

**Electrode pads and contacting procedure.** After plasma cleaning the glass substrates, UV lithography and metal evaporation are used to fabricate the Au electrodes with alignment markers. Then, the nanowires are randomly dropcast on the substrates by transferring them from the arrays with an area of $200\times200$ μm² via a pipette in ethanol onto the substrates. To contact the single nanowires to the Au electrodes, electron beam lithography and metal evaporation are used. To allow good Ohmic contact and prevent extraction barriers, the exact placement of the contact on the highly doped nanowire end segments is a crucial step.

The metals used to contact the single nanowires are Ti (200 nm) and Au (30 nm) for the electron contact (on the $n^{--}$ doped part) and Cr (3 nm), Zn (15 nm), Au (215 nm) for the hole contact (on the $p^{++}$-doped part). Before metal evaporation, the protective 50 nm $SiO_2$ shell and the native oxide of the InP are removed by etching the exposed and developed substrates in buffered HF (1:7, HF (49%):$NH_4F$ (40%)) for 10. The removal of the native oxide is a crucial step, as the latter can cause Fermi level pinning under the conduction band and hence create extraction barriers for the hole contact[67]. The freshly HF-etched samples are transferred rapidly into the evaporation chamber to minimize the regrowth of the native oxide as much as possible. To diffuse Zn into the p-type InP nanowire and create a highly p-doped layer, an additional annealing step at high temperatures has

been reported before[68]. However, we found this treatment to be damaging to our nanowires as was indicated by a strong decrease in photoluminescence efficiency. Therefore we omit this step, as the in situ doping of our nanowires and the HF etching allow to form Ohmic contacts even without annealing. Nevertheless, we evaporate Zn for the hole contact to prevent possible diffusion of Zn from the nanowire into the contact metal at elevated temperatures during the evaporation and lift-off steps. To form the electron contact, Ti and Au are evaporated with an electron beam evaporator at a pressure of $10^{-6}$ mbar at an evaporation rate of 0.3 to 2 Å s$^{-1}$ and acceleration voltage of 10 keV. For the hole contact Cr, Zn, and Au are evaporated with a thermal evaporator at $2\times10^{-6}$ mbar at a rate of 0.2 to 1.5 Å s$^{-1}$. To fabricate the resist windows, a third electron beam lithography step is added.

**Solar simulator measurements**. The $I-V$ traces are measured by illuminating the samples via a solar simulator (Oriel SOL2 94062A (6 × 6) Class ABA, Newport) with the AM1.5G spectrum at 1 sun (100 mW cm$^{-2}$) intensity. The temperature of substrates is ca. 50 °C and a silicon reference cell is used to adjust the lamp intensity of the solar simulator. Electrical probes connected to a source-measure unit (Agilent B2910) are used to contact the contact pads on the glass substrate. While measuring the current, the voltage is scanned between −1 V and 1 V in 2001 steps.

**HF etching and MoO$_X$ evaporation**. The exposed nanowire parts were etched for 10 s in buffered HF (1:7, HF (49%):NH$_4$F (40%)) to remove the protective SiO$_2$ shell (50 nm) and the native oxide of the InP under the contact. Afterwards they were immediately characterized (1 min). For the MoO$_X$ treatment, first the HF etching has been repeated after which the samples have been loaded into a thermal evaporator as quickly as possible (10 min air exposure after etching). MoO$_X$ (MoO$_3$, 99.97% trace metal basis, purchased from Sigma Aldrich) was thermally evaporated from an Mo boat at a rate of around 0.3 to 1 Å s$^{-1}$ and a pressure of $2\times10^{-6}$ mbar. The target thickness was 15 nm; however, after a short but very high spike in the evaporation rate, the final thickness was about 20 nm.

We note that the time between HF treatment and characterization was less than 1 min for the samples that have only been exposed to HF, while the time between HF treatment and pumping down for the case of additional MoO$_X$ deposition was about 10 min. This difference originates from practical constraints: HF treatment to test surface passivation was done for a single chip at a time in an adjacent lab, while in the case of MoO$_X$ deposition the etching/rinsing/drying was done in a serial manner for ten chips, which were then all carried to another part of the building, attached to a sample holder and loaded into the prepared vacuum chamber (10 min).

**Data availability**. All relevant data are available from the authors upon request.

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

## Acknowledgements

The work at AMOLF is funded by the "Nederlandse Organisatie voor Wetenschappelijk Onderzoek" (NWO) by the NWO VIDI grant (project number 14846) and by the European Research Council (Grant Agreement No. 337328).

## Author contributions

S.Z.O. and E.C.G. developed the concept and designed the experiment. S.Z.O. and H.S. performed the device fabrication and characterization under the supervision of E.C.G. A. C. grew the nanowires under the supervision of J.E.M.H. and E.P.A.M.B. S.Z.O. and E.C. G. wrote the manuscript with contributions from all authors.

## Additional information

**Competing interests:** The authors declare no competing interests.

