## [Peer Review File · Nature Communications]

Reviewers' Comments:

Reviewer #1:

Remarks to the Author:

The authors report a clever design for significantly improving the open-circuit voltage in nanowire solar cells by engineering of the nanowire surface local to the hole (p-type) ohmic contact. The implementation of the process is rather simple, it just requires patterned removal of the silicon & native oxides and thermal evaporation of molybdenum oxide, making it a process that could realistically be adapted to practical devices. The concept is really nice scientifically as well. Essentially, the band-structure effects mimic those of traditional heterojunction contact solar cells, but the differing topology of the nanowire means that the actual carrier path to the contact matches that of a homojunction cell. As a result you get carrier selectivity without the adverse cost of increased cell impedance. This is one of those nice and rare cases in physics where you 'get to have your cake and eat it too', so to speak, arising entirely from the unique topology of a nanowire compared to traditional planar electronics. I can see this inspiring quite a bit of follow-up work. I certainly think it's at the level expected for a paper in Nature Communications, but it would be good to see a few minor details addressed prior to publication.

1. Figure 1 is nice but the cross-section schematic in the top-right corner is confusing as presented. When you look at the schematic, the immediate question is: How can there be conduction and valence bands out in the vacuum outside the nanowire and MoOx coating? The cause of the confusion is evident when you eventually look at Fig. S3 – it's just that the schematic is stuffed into so small a space that the layers end before the band drawings do. Perhaps a better arrangement can be made here?

2. The schematic in Fig. 1(c) also shows an equivalent approach deployed schematically on the n-contact side. Have the authors attempted this also? If so, why isn't it also discussed here? If not, why not? I would guess a good candidate there could be ZnO? Thermal evaporation of Zn and an oxidation anneal is probably a bad idea given RTA wrecks the contacts here. But there are electrodeposition techniques for ZnO. I think at least the authors should give a little bit of discussion on this issue, it'll increase the impact of the work.

3. No explanation for the SiO₂ shell is given. Why is that added if that always needs to be removed for the other components of the device? Probably only needs one sentence, but it's a helpful detail for a broader non-expert audience.

4. Looking at the fabrication route in the SI and the schematic in Fig. 2(a) it's clear that the contacts and the MoOx layer are not fully concentric. Clearly this doesn't compromise the overall effect, looking at the data, but it must mean your devices are a little off optimum? Is it possible carrier selectivity isn't as good on the bottom surface of the nanowire as it is on the top surface, for example? It would be good to discuss this briefly and it can easily be spun in a positive way: Clearly you can get better performance from this idea than the simple prototype shown, so a reader should consider the results representative and optimisable rather than the upper bound.

5. It is interesting that in Fig. 2(b) you see a gain in both short-circuit current and open-circuit voltage, i.e., true improved efficiency, whereas in Fig. S2, where there's a gap between the MoOx layer and contact, you see the enhanced Voc come at the expense of significant Isc loss. This isn't really commented on in the current draft, where the focus is solely on Voc with no mention of Isc. I think this also warrants some brief mention. The gap doesn't really kill your effect, as you say, but it's not as though you can realistically live with that gap in practical devices, as it hurts you in other places.

6. I think some additional comment regarding the evaporation process for the MoOx should be given in the SI. What exactly is the source material (composition, purity, supplier)? Is there anything particular about the boat used, or is it just a Mo or W boat, or filament, or crucible... You

want people to use this technique for further work, right, it's how you generate more (non-self-)citations more quickly, the more you can help them to do that, the better your returns on the work.

Reviewer #2:

Remarks to the Author:

The Authors propose the deposition of a MoOx layer to create a locally induced P+ layer around the hole contact of an InP nanowire solar cell, improving its performance. However, I find that the authors do not present a compelling case and that there are several open questions throughout the paper which reduce the impact of this publication. More importantly, the device performance is poor, especially for a III-V cell. In my opinion, the work does not represent a major advance in the field and I cannot recommend its publication.

Major comments:

No direct numerical comparison is made to the state of the art InP nanowire performance. The Authors state that "the Voc values attained match state-of-the-art performance" this should be confirmed with numbers from state of the art nanowire devices.

The use of terms like 'surface heterojunction gate control' might confuse some readers. The authors indicate that no current is collected through MoOx layer and hence it is confusing to refer to it as a heterojunction.

The Authors emphasize the role of the MoOx layer in improving the hole extraction capability by creating a localized p+ layer near the hole contact. Could the increase in Voc instead be caused by an improvement in surface passivation of the InP nanowire from the MoOx layer. The enhancement in Voc despite a discontinuity in the MoOx region and the hole contact shown in the supporting information further supports this. Did the Authors try applying a MoOx layer to the entire exposed surface of the nanowire? Or perhaps just on the n-side to test their theory?

Alternatively, to test the presence of the induced p+ layer, the Authors could measure the change in resistance for a nanowire with two p-type ohmic contacts before and after MoOx deposition.

The use of HF treatment to test the effects of surface passivation is difficult, as the Authors point out due to the fast reduction in performance with air exposure time. Can the Authors comment on the time between HF treatment and measurement? Can they eliminate the possibility that the MoOx treated samples have a higher Voc because of a small time between HF treatment and MoOx evaporation?

From IV curves in Figure 2b, it appears that a Schottky barrier forms after MoOx deposition, can the Author provide an explanation of this?

The difficulty of the trade-off between thickness and surface passivation for bulk semiconductor solar cells is over-emphasized, as evidenced by industrial GaAs and Si heterojunction solar cells with efficiencies close to their theoretical limits.

Minor comments:

The Authors utilize the nomenclature "n-", this may be confusing to some readers as it appears to indicate very low n-type doping.

Caption of Figure 1 appears to be cut off

Figure 2c does not appear to be a fair comparison. The 'HF only' devices we sampled from starting Voc between 300-800, whereas the MoOx treated devices were sampled from devices with starting Voc's above 450 mV.

In figure 2a, the picture shows a layer on top of the MoOx, can the Authors explain what this layer is?

Reviewer #3:

Remarks to the Author:

The authors present an interesting study of using MoOx to affect the electrical behavior of InP

nanowires. However, there are some technical details that are necessary to clarify before I can make a recommendation:

1. It appears that the MoOx affects the effective doping in the nanowire, basically enhancing the p-doping on the p++ side. It would be very interesting to see EBIC measurements before and after the application of the MoOx to study how the pn-junction has changed.
2. In Figure 2b, the short-circuit current increases by ~100% after the application of the MoOx. However, in Figure S2, the short-circuit current appears to decrease by ~40% with a disconnected MoOx pad. Could the authors clarify the origin of this increase vs. decrease?
3. The wires show an Voc of about 0.4 to 0.8 V with or without the MoOx. Such a Voc sounds low to be limited by minority carrier leakage to the ohmic contact. Could it be that the limiting factor is recombination of electrons injected into the p-segment? Perhaps the MoOx increases the effective doping on the p-side so much that electrons are not effectively injected there, but start to predominantly recombine in the junction, enhancing Voc?
4. It would be interesting to see light and dark IV curves, both with and without MoOx, to see whether the superposition principle holds for these nanowires.
5. Instead of talking about charge carrier-selective contacts, would it make sense to say that the MoOx modified the effective doping of the nanowires? Such external "doping" would definitely have strong interest for defining pn-junctions in nanowires.

Regarding the supporting information:

1. In Sample Fabrication, the authors could comment more on the crystal phase of the nanowire.
2. The nanowires are grown with selective area MOVPE. I would have expected radial growth at the same time as axial growth, which would lead to a pn-junction in the radial direction also. However, the authors present the nanowire as an axial p-i-n junction. Could the authors comment in the methods section why such radial growth apparently does not occur to disturb the radial doping profile of the nanowire?
3. Regarding the red curve in Figure S1, the authors state that the blueshift occurs because of band filling of the lower band. But that lower band would be for Wz InP? However, the red curve is taken at the n—end of the nanowire where according to TEM in Supporting Information of Ref. [1], the crystal phase is mixed. For such mixed crystal phase, would we expect the lower band from Wz InP to show up clearly in the bandstructure?
4. In the schematic in Figure S3, it looks like the surface depletion does not affect the center of the nanowire. In that case, there would be an unmodified conduction channel in the middle of the nanowires for electrons to reach the ohmic contact. I have doubts that the leakage of electrons to the metallic/ohmic contact would be reduced much by such non-complete reduction of the conduction channel size. Could the authors comment on this?

Response to reviewer comments for
“Charge Carrier-Selective Contacts for Nanowire Solar Cells”

Reviewers' comments:

Reviewer #1 (Remarks to the Author):

The authors report a clever design for significantly improving the open-circuit voltage in nanowire solar cells by engineering of the nanowire surface local to the hole (p-type) ohmic contact. The implementation of the process is rather simple, it just requires patterned removal of the silicon & native oxides and thermal evaporation of molybdenum oxide, making it a process that could realistically be adapted to practical devices. The concept is really nice scientifically as well. Essentially, the band-structure effects mimic those of traditional heterojunction contact solar cells, but the differing topology of the nanowire means that the actual carrier path to the contact matches that of a homojunction cell. As a result you get carrier selectivity without the adverse cost of increased cell impedance. This is one of those nice and rare cases in physics where you ‘get to have your cake and eat it too’, so to speak, arising entirely from the unique topology of a nanowire compared to traditional planar electronics. I can see this inspiring quite a bit of follow-up work. I certainly think it’s at the level expected for a paper in Nature Communications, but it would be good to see a few minor details addressed prior to publication.

1. Figure 1 is nice but the cross-section schematic in the top-right corner is confusing as presented. When you look at the schematic, the immediate question is: How can there be conduction and valence bands out in the vacuum outside the nanowire and MoOx coating? The cause of the confusion is evident when you eventually look at Fig. S3 – it’s just that the schematic is stuffed into so small a space that the layers end before the band drawings do. Perhaps a better arrangement can be made here?

We thank the reviewer for pointing out the flaw in Fig. 1. We have edited the figure as suggested.

2. The schematic in Fig. 1(c) also shows an equivalent approach deployed schematically on the n-contact side. Have the authors attempted this also? If so, why isn’t it also discussed here? If not, why not? I would guess a good candidate there could be ZnO? Thermal evaporation of Zn and an oxidation anneal is probably a bad idea given RTA wrecks the contacts here. But there are electrodeposition techniques for ZnO. I think at least the authors should give a little bit of discussion on this issue, it’ll increase the impact of the work.

We show the novel contacting scheme applied to both p- and n-type contacts in the schematic, as it should be possible to apply to both (just as heterojunction contacts have been applied to both in bulk solar cells, see for example references 3, 4, 8 and 15). However, we only tested the scheme on the p-type contact since for InP (and III-V materials more generally) it is particularly the p-type contact that is very challenging. For example the widely used p-type dopant Zn tends to lead to insufficient doping levels, especially for the MOCVD growth method, shows high diffusivity (especially at high concentrations) thereby leading to non-ideal doping profiles and is even suspected to induce recombination active defects. We have added a short discussion about this in the revised manuscript.

3. No explanation for the SiO₂ shell is given. Why is that added if that always needs to be removed for the other components of the device? Probably only needs one sentence, but it's a helpful detail for a broader non-expert audience.

The SiO₂ layer is used to improve the stability of the InP nanowires (we observed faster degradation of devices made without it) and to avoid clustering of the drop-cast wires. It may also provide some surface passivation, although that was not studied here. Please note that it is only removed under the contact or the surface gate layer and not entirely from the whole nanowire. We have now added a sentence to the updated draft about this.

4. Looking at the fabrication route in the SI and the schematic in Fig. 2(a) it's clear that the contacts and the MoOx layer are not fully concentric. Clearly this doesn't compromise the overall effect, looking at the data, but it must mean your devices are a little off optimum? Is it possible carrier selectivity isn't as good on the bottom surface of the nanowire as it is on the top surface, for example? It would be good to discuss this briefly and it can easily be spun in a positive way: Clearly you can get better performance from this idea than the simple prototype shown, so a reader should consider the results representative and optimisable rather than the upper bound.

We agree that by applying the MoOx layer conformally around the nanowire (as in a wrap-around gate) the performance should improve even more. We have added a small discussion about this point.

5. It is interesting that in Fig. 2(b) you see a gain in both short-circuit current and open-circuit voltage, i.e., true improved efficiency, whereas in Fig. S2, where there's a gap between the MoOx layer and contact, you see the enhanced Voc come at the expense of significant Isc loss. This isn't really commented on in the current draft, where the focus is solely on Voc with no mention of Isc. I think this also warrants some brief mention. The gap doesn't really kill your effect, as you say, but it's not as though you can realistically live with that gap in practical devices, as it hurts you in other places.

It is true that the Isc sometimes increases and sometimes decreases after application of the MoOx layer. The reason is that the MoOx is also capped by gold (in order to maintain the high MoOx work function and avoid its degradation due to ambient contaminants) which blocks a significant portion of the light. Therefore, we would actually always expect to see a decrease in Isc. For devices with a relatively good initial performance, we do always observe the expected decrease in Isc (but increase in Voc). Only in devices with relatively poor initial performance, we see that the improvement in the contact upon application of MoOx can overcome the shadowing loss. As suggested by the reviewer, we have added a short discussion about this point in the manuscript.

6. I think some additional comment regarding the evaporation process for the MoOx should be given in the SI. What exactly is the source material (composition, purity, supplier)? Is there anything particular about the boat used, or is it just a Mo or W boat, or filament, or crucible... You want people to use this technique for further work, right, it's how you generate more (non-self-)citations more quickly, the more you can help them to do that, the better your returns on the work.

We thank the reviewer for pointing out the missing information. We have added them to the SI.

Reviewer #2 (Remarks to the Author):

The Authors propose the deposition of a MoOx layer to create a locally induced P+ layer around the hole

contact of an InP nanowire solar cell, improving its performance. However, I find that the authors do not present a compelling case and that there are several open questions throughout the paper which reduce the impact of this publication. More importantly, the device performance is poor, especially for a III-V cell. In my opinion, the work does not represent a major advance in the field and I cannot recommend its publication.

We hope that the additional data added has now convinced this reviewer. We find it important to stress that these devices do not have poor performance, they actually show results comparable to record InP nanowire solar cells, both nanowire array and single nanowire measurements (see below).

Major comments:

1. No direct numerical comparison is made to the state of the art InP nanowire performance. The Authors state that “the Voc values attained match state-of-the-art performance” this should be confirmed with numbers from state of the art nanowire devices.

We apologize for this oversight. We have now cited previous work showing record results for both single nanowire InP (800 - 890 mV) and InP nanowire array (760 mV, 17.8% efficiency, and 906 mV, 13% efficiency) solar cells. We have also added these numbers to the text to help the reader make their own comparison between these values and the 835 mV Voc achieved with our MoOx surface gate.

2. The use of terms like ‘surface heterojunction gate control’ might confuse some readers. The authors indicate that no current is collected through MoOx layer and hence it is confusing to refer to it as a heterojunction.

We apologize for the confusion. The heterojunction analogy was intended to help the reader see the similarities to that commonly used contact geometry, while also providing a reference geometry to point out the important differences. To avoid confusion, we have now removed “heterojunction” and only used “surface gate” when referring to our contacting scheme.

3. The Authors emphasize the role of the MoOx layer in improving the hole extraction capability by creating a localized p+ layer near the hole contact. Could the increase in Voc instead be caused by an improvement in surface passivation of the InP nanowire from the MoOx layer. The enhancement in Voc despite a discontinuity in the MoOx region and the hole contact shown in the supporting information further supports this.

We agree that in principle improved surface passivation could explain an increase in Voc. However, there are several points that strongly suggest a largely different impact of the MoOx treatment than simple surface passivation:

1. We only see very minor absolute improvements in Voc after treatment with HF, which is however a benchmark for low surface recombination velocities in InP (see also response to reviewer #3). In fact, HF or HCL etching are being used to get InP surfaces epi-ready because they can completely remove the native oxide (see e.g. *Pluchery et al., J. Appl. Phys., Vol. 94, No. 4, 15 August 2003*). As noted by the reviewer, and as we show in Figure 3, this surface termination is not stable. However, the time between HF etching and characterization was shorter (~1 min) than for the air exposure during the MoOx treatment (<10min) (see also response to #5 below), i.e. any effect of surface passivation due to native oxide removal for the MoOx samples is likely lower than for the HF-only treated samples.

2. We observe a large spread for starting Voc values and a varying PL intensity between different and even on the same nanowires (see also PL maps in the SI). Using surface passivation as explanation, we find it difficult to explain such results. That would require postulating a large spread for the surface passivation for the same growth batch and even on the same single nanowire, even though all the nanowires were exposed to the exact same post-growth surface treatment, which has been shown by previous PL studies to lead to high surface passivation properties.
3. The spread in Voc decreases after MoOx treatment but not after HF treatment, which clearly shows that the MoOx treatment (HF + MoOx) has a distinctly different effect on the nanowires than the HF only treatment. Not only does the MoOx treatment simply increase the Voc values to a larger value than the HF treatment (see point 1) but it also changes the spread, i.e. it raises all Voc values to a new common level across devices. This strongly suggests a new mechanism limiting the Voc after the MoOx treatment compared to before.
4. The results of the suggested experiments by the referee (in #4 below), investigating the influence of the MoOx further, also confirm the surface gating hypothesis. Fig. S2a) shows how the application of MoOx on the n-type side of the single nanowire solar cell strongly degrades the performance. This would not be the case if surface passivation would be causing the strongly improved performance after MoOx application.

In summary, we find initial stochastic variations in local dopant incorporation and activation and the subsequent surface gating effect a much more likely explanation for the observed enhancement. To clarify this point we added a discussion about the matter to the manuscript and conducted additional experiments.

4. Did the Authors try applying a MoOx layer to the entire exposed surface of the nanowire? Or perhaps just on the n-side to test their theory? Alternatively, to test the presence of the induced p+ layer, the Authors could measure the change in resistance for a nanowire with two p-type ohmic contacts before and after MoOx deposition.

We thank the referee for suggesting those additional and clarifying experiments. We have now added new data in Figure S2. The results clearly show that applying MoOx to the n-side of the nanowire degrades the solar cell performance strongly (Fig. S2a), and that the resistance in p-type InP nanowires is reduced after applying a MoOx surface gate on the center of the wires. Those experiments strongly support our surface gating hypothesis against other explanations, such as surface passivation. We have included the data in the SI and added additional discussion to the main text.

5. The use of HF treatment to test the effects of surface passivation is difficult, as the Authors point out due to the fast reduction in performance with air exposure time. Can the Authors comment on the time between HF treatment and measurement? Can they eliminate the possibility that the MoOx treated samples have a higher Voc because of a small time between HF treatment and MoOx evaporation?

We thank the referee for noting this important point. We can be completely certain that the higher Voc is not due to a shorter air exposure; it occurs in spite of a longer air exposure. The time between HF treatment and solar cell testing was less than 1 minute while the time between HF treatment and pumping down for MoOx deposition was 5-10 minutes. This difference came entirely from practical

constraints: HF treatment to test surface passivation was done for a single chip at a time in a lab adjacent to the characterization setup, while in the case of MoOx deposition the etching/rinsing/drying was done in a serial manner for 10 chips, which were then all carried to another part of the building, attached to a sample holder and loaded into the prepared vacuum chamber (<10 min). We have added this important point to the manuscript.

6. From IV curves in Figure 2b, it appears that a Schottky barrier forms after MoOx deposition, can the Author provide an explanation of this?

We ascribe the extraction barrier to a non-ideal effective doping profile along the hole extraction path. It is important to note that we also observe extraction barriers before the MoOx surface treatment (see e.g. Figure S3).

For the surface modified nanowires, holes first encounter the high effective doping concentration under the MoOx surface gate, i.e. the very region which is responsible for the studied increase in carrier selectivity of the contact by suppressing the electron current into the same direction. However, after traversing the surface gated region in longitudinal nanowire direction the holes encounter a region with the as-grown carrier concentration under the ohmic metal contact. This effective concentration profile, metal|n⁺|n|p⁺|p|metal is a non-ideal situation (the ideal case would be metal|n⁺|n|p|p⁺|metal) and can explain the occurrence of extraction barriers. This is further supported by the fact that devices with a gap between the MoOx pad and the metal contact (many of the devices) typically had larger extraction barriers. Furthermore, the 100 nm thick Au pad to prevent ambient contaminants to reduce the MoOx work function shadows a large fraction of the wire from light illumination.

All of those issues mentioned here are related to the specific horizontal single nanowire device geometry used to study the effect. We stress that with an ideal transparent wrap-around geometry and ohmic contact formation only on the very nanowire tip, as would be possible in a vertical nanowire array geometry, those issues would be prevented.

To clarify this point we have added this discussion to the SI (and a respective reference in the text).

7. The difficulty of the trade-off between thickness and surface passivation for bulk semiconductor solar cells is over-emphasized, as evidenced by industrial GaAs and Si heterojunction solar cells with efficiencies close to their theoretical limits.

We agree that in the case of intrinsic hydrogenated amorphous silicon (a-Si:H(i)) it is possible to reach very high open-circuit voltages for silicon heterojunction solar cells even for thin 5-10 nm layers. The current record holder, the Kaneka HIT cell (26.7%) reaches V_{OC} values of 738 mV (*M. Green et al., Prog. Photovolt. Res. Appl* 25, 668–676 (2017), <https://doi.org/10.1002/pip.2909>), only 14 mV less than the theoretical Auger limit of 752 mV (for a thickness of 165 μm and resistivity of 3 Ωcm^2) (Yoshikawa et al., *Nat. Energy* 2, 17032, <https://doi.org/10.1038/nenergy.2017.32>). However, the record silicon heterojunction solar cell is still about 10% (rel.) below the absolute theoretical maximum efficiency for silicon solar cells (29.4 %), partially due to imperfect contact and surface passivation (Yoshikawa et al., *Nat. Energy* 2, 17032, <https://doi.org/10.1038/nenergy.2017.32>). Thicker a-Si:H(i) layers can give superior surface passivation (e.g. Fig. 4 in *Mueller et al., Energy Procedia* 15, 97 (2012) <https://doi.org/10.1016/j.egypro.2012.02.012>) and hence a higher V_{OC} , but the FF and J_{SC} suffer due to an increase in resistance and parasitic absorption (e.g. Figure 10 in *Z.C. Holman et al. IEEE Journal of Photovoltaics* 2, 7-15 (2012), [10.1109/JPHOTOV.2011.2174967](https://doi.org/10.1109/JPHOTOV.2011.2174967)). Therefore, the current high efficiency solar cells have already optimized this trade-off, which is also indicated by the fact that the current

Kaneka record HIT cell (26.7%) has a V_{OC} of 738 mV that is in fact lower than the V_{OC} of 750 mV of a previous record holder, the Panasonic HIT cell (24.7%) (M. Taguchi et al., *IEEE Journal of Photovoltaics* 4, 96-99 (2013), [10.1109/JPHOTOV.2013.2282737](https://doi.org/10.1109/JPHOTOV.2013.2282737)).

Additionally, and related to the above mentioned thickness trade-off, very thin intrinsic (< 3nm) a-Si:H layers lead to a loss of passivation during the subsequent plasma-deposited doped a-Si:H emitter or BSF layer (see e.g. Fig. 1 in S. de Wolf and G. Beaucarne, *Appl. Phys. Lett.* 88, 022104 (2006), <https://doi.org/10.1063/1.2164902>). Furthermore, we note that the thickness trade-off is even more severe for other passivation layers. The dielectric Al_2O_3 can only be used as very thin (1-2 nm) tunnel layer, e.g. in MIS-type junctions (e.g. A. Richter et al. *Progress in Photovoltaics*, 1-8 (2017), <https://doi.org/10.1002/pip.2960>), in which case the passivation properties are largely degraded (e.g. Fig. 6 in J. Schmidt et al., 35th *IEEE Photovoltaic Specialists Conference* (2010), [10.1109/PVSC.2010.5614132](https://doi.org/10.1109/PVSC.2010.5614132)) or with thicker layers and improved passivation in local point contact schemes (e.g. A. Richter et al. *Progress in Photovoltaics*, 1-8 (2017), <https://doi.org/10.1002/pip.2960>). In summary, if the thickness trade-off can be removed, potentially even higher efficiencies can be expected.

We have added a few sentences in the introduction to make clear that silicon heterojunction have indeed reached very high open-circuit voltages by making use of the remarkable passivation properties of a-Si:H(i) layers (even when used with 5-10 nm thicknesses).

Minor comments:

8. The Authors utilize the nomenclature “n-”, this may be confusing to some readers as it appears to indicate very low n-type doping.

We thank the reviewer for pointing out this source of possible confusion. We have changed the nomenclature to the suggested “n⁺” in the text and all respective figures.

9. Caption of Figure 1 appears to be cut off.

We have fixed this issue.

10. Figure 2c does not appear to be a fair comparison. The ‘HF only’ devices we sampled from starting V_{oc} between 300-800, whereas the MoOx treated devices were sampled from devices with starting V_{oc} 's above 450 mV.

The referee’s perception of the visual overemphasis of the superiority of the HF+MoOx treatment over the actual advantage stems from the difference in numbers of measured devices. In the current version of the Figure, we show the results of 8 devices for the HF only treatment and of 6 devices for the HF+MoOx treatment. Indeed the three lowest initial V_{OC} values were obtained for the “HF only” batch. We could delete the two lowest initial V_{OC} values (thereby having 6 data points for each treatment), which however would not change the clear message of this Figure. As also mentioned by the reviewer, the HF+MoOx treatment is actually advantageous over the “HF only” treatment. Because of the generally low numbers of measured devices (due to the contacting related issues as described in the SI) we prefer not to eliminate 2 out of 8 data points for the HF only treatment and would prefer to keep Figure 2c unchanged.

11. In figure 2a, the picture shows a layer on top of the MoOx, can the Authors explain what this layer is?

We apologize for this omission in the labelling. The layer is an Au cap used to prevent the MoOx work function from changing due to ambient contaminants. We have edited the Figure.

Reviewer #3 (Remarks to the Author):

The authors present an interesting study of using MoOx to affect the electrical behavior of InP nanowires. However, there are some technical details that are necessary to clarify before I can make a recommendation:

1. It appears that the MoOx affects the effective doping in the nanowire, basically enhancing the p-doping on the p++ side. It would be very interesting to see EBIC measurements before and after the application of the MoOx to study how the pn-junction has changed.

We agree with the reviewer in principle. Studying the exact change in location of the pn-junction, or rather the i-p junction (as the doping profile is n-i-p), would be very insightful. Unfortunately, our device geometry does not allow such a localization via EBIC unhindered. The MoOx layer, which enhances the p-type doping, is covered by a 50-100 nm Au layer to prevent ambient contaminants from lowering the work function. This layer will strongly scatter electrons before they can even be absorbed and induce current in the InP nanowire. To overcome this problem, we attempted to fabricate devices without the additional Au layer, but the results showed no clear difference in performance of the wire to the initial case without MoOx. The high work function of MoOx must be very sensitive to surface contaminants or even changes in the oxidation state, as was also reported in the literature. Another serious issue is related to the electron beam-induced damage we also observed for our EBIC measurements shown in previous work (doi: 10.1038/nano.2016.162). Those measurements were done in a “single-snapshot-mode”, because we observed rapid degradation of the current collection during exposure. Therefore, EBIC measurements before and after the treatment of the same wire would be highly desirable, but are unlikely to yield the required high quality EBIC images that are needed to clearly locate the junction. Last but not least, we do not expect EBIC profiles to clearly show that the current collection takes place via the metal contacts instead of the MoOx pad (which we can rule out due to our other results). Charge carriers that are excited by the electron beam next to the MoOx-Au pad will result in detection of current (under short-current conditions. Those are essentially the conditions when the charge carriers are photo-excited (due to the optical opaque Au protection layer). As a result, even if the MoOx pad would extract current (which it doesn't as we discuss in the paper), the profile would be essentially the same as for the case of current collection via the metal contacts at the nanowire tips (falling off towards the center of the beam exposed part due to a finite diffusion length). To conclude, we see currently too many obstacles for the likelihood of a successful EBIC experiment.

2. In Figure 2b, the short-circuit current increases by ~100% after the application of the MoOx. However, in Figure S2, the short-circuit current appears to decrease by ~40% with a disconnected MoOx pad. Could the authors clarify the origin of this increase vs. decrease?

Please see response to reviewer #1, comment 5.

3. The wires show an Voc of about 0.4 to 0.8 V with or without the MoOx. Such a Voc sounds low to be

limited by minority carrier leakage to the ohmic contact. Could it be that the limiting factor is recombination of electrons injected into the p-segment? Perhaps the MoOx increases the effective doping on the p-side so much that electrons are not effectively injected there, but start to predominantly recombine in the junction, enhancing Voc?

We agree completely with the reviewer's suggestion that the MoOx increases the effective doping on the p-side so much that the electrons are not injected there. This is precisely the mechanism for reducing minority carrier leakage to the ohmic contact (see also response to comment 5) We have modified the text to make this more clear.

4. It would be interesting to see light and dark IV curves, both with and without MoOx, to see whether the superposition principle holds for these nanowires.

We agree that this data should have been included in the first place. It has now been added to the SI and is referenced in the main text. Interestingly it shows, that the nanowires are very resistive in the dark, that is a simple negative shift of a typical dark diode I-V curve does not occur. We ascribe this effect to the non-ideal doping profile and the overall lengths of the nanowires (with an intrinsic and a rather large unintended low p-type segment). As we now also discuss in the SI and the main text, we believe that the resistive behavior in the dark might be linked to the occurrence of charge carrier extraction barriers.

5. Instead of talking about charge carrier-selective contacts, would it make sense to say that the MoOx modified the effective doping of the nanowires? Such external "doping" would definitely have strong interest for defining pn-junctions in nanowires.

We apologize that this was not clear; the modified effective doping by surface gating with the MoOx layer is the mechanism for the improvement in the p-type charge carrier-selective contact. We have altered the text to make this more apparent.

Regarding the supporting information:

1. In Sample Fabrication, the authors could comment more on the crystal phase of the nanowire.

Additional discussion about the crystal phase has been added.

2. The nanowires are grown with selective area MOVPE. I would have expected radial growth at the same time as axial growth, which would lead to a pn-junction in the radial direction also. However, the authors present the nanowire as an axial p-i-n junction. Could the authors comment in the methods section why such radial growth apparently does not occur to disturb the radial doping profile of the nanowire?

We agree with the reviewer that selective area MOVPE does not necessarily lead to axial nanowire growth. To grow nanowire structures, as done in this work, the growth has to be selective for the top [111]A surface over the {110} side surfaces. However, for the conditions chosen, 730 °C and a pitch < 1000 nm, we observed negligible growth on the side facets when studied in the TEM. We note that to our knowledge, the detailed theoretical description of the growth mechanism depending on pitch, nanowire diameter, length, dopants and other parameters is currently very limited (*Q. Gao et al., Nano Lett.* 16, 4361-4367 (2016), <http://dx.doi.org/10.1021/acs.nanolett.6b01461>). Importantly, our results

indicate the absence of any pronounced effect of a possible core-shell structure: (I) The nanowires show rectifying photovoltaic behavior as expected for a p-i-n structure when contacted with the positive pole on the p-type side and the negative one on the n-type side. Under illumination, the holes (electrons) are driven towards the p-type (n-type) side, resulting in a negative photocurrent (against the forward diode direction). (II) HF not only etches the native oxide but also InP itself. Therefore, even if a thin (1-2 nm) shell exists initially, the HF treatment for the HF - only and the HF + MoOx + Au treatment is likely to remove this shell. (III) The duration of HF etching is the same for the HF-only and the HF + MoOx + Au treatment. Therefore, the effect of the MoOx is still clearly distinct over the HF treatment alone. (IV) The results of our photoluminescence measurements, the application of MoOx on symmetric p+pp+ wires and on the n-type part of p-i-n wires (as requested by reviewer #2) are consistent with the assumption of an axial doping geometry, too. We added this discussion to the Methods section of the SI to clarify this point.

3. Regarding the red curve in Figure S1, the authors state that the blueshift occurs because of band filling of the lower band. But that lower band would be for Wz InP? However, the red curve is taken at the n— end of the nanowire where according to TEM in Supporting Information of Ref. [1], the crystal phase is mixed. For such mixed crystal phase, would we expect the lower band from Wz InP to show up clearly in the bandstructure?

We thank the reviewer for pointing out the importance of the exact crystal structure of the highly doped n⁺⁺ end part. We have added two sentences to point this out.

4. In the schematic in Figure S3, it looks like the surface depletion does not affect the center of the nanowire. In that case, there would be an unmodified conduction channel in the middle of the nanowires for electrons to reach the ohmic contact. I have doubts that the leakage of electrons to the metallic/ohmic contact would be reduced much by such non-complete reduction of the conduction channel size. Could the authors comment on this?

We agree partially. In order to see the strong effects we observe, the nanowire should show at least some increase in carrier density throughout the whole cross-section, although this may not be uniform due to the non-uniform surface gate. We have added additional discussion about this point.

Reviewers' Comments:

Reviewer #1:

Remarks to the Author:

The authors have made significant revisions in response to the referees' comments. I find these appropriate and compelling. I would be happy to see the paper published in its current form.

Reviewer #2:

None

Reviewer #3:

Remarks to the Author:

Considering the additional data, the revisions in the main text and supplementary information, and the replies to the questions raised previously by the reviewers, I believe that the study is now of interest for the readership of the journal.